# Preparation of a Fucoidan-Grafted Hyaluronan Composite Hydrogel for the Induction of Osteoblast Differentiation in Osteoblast-Like Cells

**DOI:** 10.3390/ma14051168

**Published:** 2021-03-02

**Authors:** Fu-Yin Hsu, Jheng-Jie Chen, Wen-Chieh Sung, Pai-An Hwang

**Affiliations:** 1Department of Bioscience and Biotechnology, National Taiwan Ocean University, Keelung City 20224, Taiwan; fyhsu@mail.ntou.edu.tw (F.-Y.H.); bravoa33049@gmail.com (J.-J.C.); 2Department of Food Science, National Taiwan Ocean University, Keelung City 20224, Taiwan; sungwill@mail.ntou.edu.tw; 3Center of Excellence for the Oceans, National Taiwan Ocean University, Keelung City 20224, Taiwan

**Keywords:** fucoidan, hyaluronan, hydrogel, osteoblast

## Abstract

A suitable bone substitute is necessary in bone regenerative medicine. Hyaluronan (HA) has excellent biocompatibility and biodegradability and is widely used in tissue engineering. Additionally, research on fucoidan (Fu), a fucose- and sulfate-rich polysaccharide from brown seaweed, for the promotion of bone osteogenic differentiation has increased exponentially. In this study, HA and Fu were functionalized by grafting methacrylic groups onto the backbone of the chain. Methacrylate-hyaluronan (MHA) and methacrylate-fucoidan (MFu) were characterized by FTIR and ^1^H NMR spectroscopy to confirm functionalization. The degrees of methacrylation (DMs) of MHA and MFu were 9.2% and 98.6%, respectively. Furthermore, we evaluated the mechanical properties of the hydrogels formed from mixtures of photo-crosslinkable MHA (1%) with varying concentrations of MFu (0%, 0.5%, and 1%). There were no changes in the hardness values of the hydrogels, but the elastic modulus decreased upon the addition of MFu, and these mechanical properties were not significantly different with or without preosteoblastic MG63 cell culture for up to 28 days. Furthermore, the cell morphologies and viabilities were not significantly different after culture with the MHA, MHA-MFu0.5, or MHA-MFu1.0 hydrogels, but the specific activity and mineralization of alkaline phosphatase (ALP) were significantly higher in the MHA-MFu1.0 hydrogel group compared to the other hydrogels. Hence, MHA-MFu composite hydrogels are potential bone graft materials that can provide a flexible structure and favorable niche for inducing bone osteogenic differentiation.

## 1. Introduction

Traumatic brain injury (TBI) is a life-threatening condition characterized by internal brain swelling, the degree of which can vary greatly. Currently, treatment of TBI consists mainly of two-stage surgical procedures, decompressive craniectomy (DC) and cranioplasty [1]. In the first stage, DC is conducted to remove a large portion of the calvarial bone to allow unimpeded brain swelling, which is considered to be a critical defect, as the bone will not naturally heal by itself. After brain swelling has subsided, the second stage, termed cranioplasty, is performed to close the cranial vault [2]. The average time between DC and cranioplasty has been reported to be up to several weeks; during this time, the craniectomy site typically develops a more concave shape, and there are functional and neurologic declines that are independent of the original injury. These declines have variably been termed the syndrome of the trephined (SoT), which is a severe neurological condition characterized by mood swings, fatigue, seizures, problems with motor skills, and concentration issues occurring at an average of five months following DC [3]. In addition, hydrocephalus or subdural effusion may form during the latter stage of DC, and additional shunt surgery is required [4], which is one of the reasons for delaying the course of cranioplasty treatment. Following the DC stage, materials that are utilized for cranial reconstruction include autografts, allografts, polymethylmethacrylate (PMMA), polyether ether ketone (PEEK), hydroxyapatite, and titanium in cranioplasty [5].

The current two-stage surgical treatment is disadvantageous because it requires two separate surgeries, extends the patient’s recovery time, and may lead to the neurological disease SoT. To cope with various changes in the different periods after TBI, cranial repair surgery needs a suitable scaffold that can be used to treat TBI in a single-stage surgery, which can maintain flexibility during brain swelling and then change into bone tissue after brain swelling has subsided [6].

Hydrogels are a three-dimensional (3D) network of hydrophilic polymers that are able to absorb large amounts of water or biological fluids [7]. Hydrogels have a high degree of flexibility that is similar to natural tissue due to their large water content. Hydrogels mimic the complexity of the extracellular microenvironment and have been recommended to overcome the limitations of two-dimensional (2D) culture platforms. Hydrogels can be engineered to recapitulate fundamental parameters of the extracellular matrix (ECM), such as stiffness, porosity, and hydration, which are able to act as mechanotransductors to influence cell differentiation [8]. Therefore, hydrogels have been increasingly used in tissue regeneration, as they can provide scaffolds with excellent biodegradability and biocompatibility [9]. Hyaluronan (HA), [α-1,4-D-glucuronic acid-β-1,3-N-acetyl-D-glucosamine] is a naturally occurring hydrophilic high-molecular-weight amine polymer sugar. It is a polymer found abundantly in the ECM that provides mechanical support [10]. HA is a natural biomaterial commonly used as a hydrogel matrix in bone tissue engineering. It can absorb large amounts of water and has adjustable physical properties, uniform cell distribution, high permeability, and metabolic waste [11,12]. Due to its disadvantages of hydrophilicity and lack of mechanical integrity, HA needs to be chemically modified and crosslinked for use in bone tissue engineering applications [13,14]. Methacrylated HA (MHA) hydrogels with the addition of a photoinitiator can polymerize for network formation upon UV exposure and result in network formation, leading to increased rigidity and therefore better resistance against degradation [15,16]. However, HA only supports the attachment and proliferation of osteoblast cells but does not support cell mineralization [17,18]. Therefore, remodeling MHA hydrogels by incorporating components is needed to induce osteogenesis [19,20].

Methacrylated chondroitin sulfate (MCS) has been combined with MHA to yield hydrogels. Guo et al. [21] reported that collagen/MHA/MSC mixed hydrogel was efficient in chondrocytes secreting glycosaminoglycan and collagen II, and Costantini et al. [22] also reported that methacrylated gelatin (MGel)/MHA/MSC mixed hydrogels enhanced the viability and chondrogenic differentiation of bone marrow-derived human mesenchymal stem cells (BM-MSCs). However, the natural sulfated polymer, fucoidan (Fu), has not been mixed with MHA hydrogels. Fu is a class of sulfated, fucosylated polysaccharides found in brown seaweed, identified by Kylin [23]. They have a backbone made of α(1→3)-l-fucopyranose residues or alternating α(1→3) and α(1→4)-linked l-fucopyranosyl residues, and both forms may be sulfate-substituted [24]. The Fu biological activity intensity varies with species, route of extraction, composition, structure, and molecular weight [25]. It has been reported that Fu can promote osteoblast differentiation via JNK- and ERK-dependent BMP2-Smad 1/5/8 signaling [26], and our previous study demonstrated that Fu promotes bone osteogenic differentiation properties [27], which is important regarding the pharmaceutical and biomedical applications of this polysaccharide. Currently, only a few studies have used Fu as a hydrogel structure and support material [28,29,30]. To our knowledge, the role of Fu in hydrogels to induce bone osteogenic differentiation has not yet been addressed.

In this study, we established a methodology for the preparation of Fu-mixed HA photo-crosslinked structures to provide a favorable niche to induce bone osteogenic differentiation. We hypothesized that the methacrylated-Fu (MFu)-MHA photo-crosslinked hydrogel could support osteogenic differentiation and mineralization of osteoblasts for use in bone tissue engineering. We tested this hypothesis by evaluating the effects of MFu-MHA on the hydrogel mechanical properties, cell proliferation, cell attachment, alkaline phosphatase (ALP) production, and mineralization and compared these effects with those of MHA hydrogels.

## 2. Materials and Methods

### 2.1. Materials

HA (sodium salt form, 620–1200 kDa) was purchased from NovaMatrix (Sandvika, Norway). Fu was purified using protocols based on previous methods [31] in which *Laminaria japonica* was treated with distilled water and boiled at 100 °C for 30 min. The supernatant was added with 4 M CaCl_2_ to precipitate alginic acid and recentrifuged at 10,000× *g* for 20 min. The polysaccharides were dialyzed by molecular weight cutoff membrane (10 kDa, Thermo Fisher Scientific Inc., Waltham, MA, USA) with deionized water for 48 h. A 3 times volume of ethanol was then added to precipitate crude fucoidan. The crude fucoidan was fractionated through DEAE-Sephadex A-25 chromatography (Cl-form, Pharmacia, Uppsala, Sweden) with 1.5 M NaCl elution. The final fucoidan sample had an average molecular weight of 35–40 kDa [32], 214.36 ± 0.29 μmol fucose [33], and 30.4 ± 2.8% sulfate [34].

### 2.2. Preparation MHA and MFu

MHA was modified following an adapted protocol by Leach et al. [35]. HA (1.0 g) was dissolved in 100 mL of reverse osmosis (RO) water and stirred overnight at 5 °C for complete dissolution. Subsequently, dimethylformamide (DMF) (BioSolve, Valkenswaard, the Netherlands) was added dropwise to obtain a DMF/water ratio of 2/3 (*v*/*v*). Then, 12 mL of a solution of methacrylic anhydride (MA) in a DMF/water solution (1/1 *v*/*v*) was added while maintaining the pH at 8 using NaOH (0.5 M). After overnight stirring at 5 °C, the solution was precipitated with 2 times the volume of ethanol, and the precipitate was dissolved in distilled water and transferred to a dialysis membrane (MWCO 12–14 kDa, Medicell, London, UK) and dialyzed for 5 days at 4 °C against RO water. The remaining MHA solution was freeze-dried and stored at −20 °C until characterization and use.

MFu was modified following an adapted protocol by Reys et al. [30]. Fu (1.0 g) was dissolved in 100 mL of RO water and stirred overnight at 50 °C for complete dissolution. MA (12 mL) in a DMF/water solution was added while maintaining the pH at 8 using NaOH (0.5 M). After 6 h of stirring at 50 °C, the solution was precipitated with 2 times the volume of ethanol, and the precipitate was dissolved in distilled water, transferred to a dialysis membrane (MWCO 12–14 kDa), and dialyzed for 5 days at 4 °C against RO water. The remaining MFu solution was freeze-dried and stored at −20 °C until characterization and use.

### 2.3. Characterization of MHA and MFu Using Fourier Transform Infrared Spectroscopy (FTIR) and ^1^H NMR Spectroscopy

The molecular structures of HA, MHA, Fu, and MFu were investigated using an MIDAC 2000 FTIR (MIDAC Corporation, Costa Mesa, CA, USA). The powdered samples were mixed with potassium bromide (KBr) and processed into pellets, and the FTIR spectra were recorded between 400 and 2000 cm^−1^. The degree of methacrylation (DM) of MHA and MFu was quantified by ^1^H NMR spectroscopy. The ^1^H NMR spectra of HA, MHA, Fu, and MFu were collected in deuterated water (D_2_O) at a concentration of 5 mg/mL at 50 °C and recorded on Bruker Advance III with the following spectral conditions: 300 Hz spectra with 90° impulses and 4 s of acquisition time.

DM was calculated by comparing the two methylene (=CH_2_) protons of the MA group at δ–6.2 ppm and δ–5.8 ppm [36] to three methyl (−CH_3_) protons (δ–2.1 ppm) of HA [36], and six methyl (−CH_3_) protons (δ–1.3 ppm) of Fu [28]. *I*_methylene_ and *I*_CH3_ correspond to the relative integration of methylene protons of MA and methyl protons of HA or Fu. *N_methylene_* and *N*_CH3_ correspond to the number of protons in methylene of MA and methyl of HA or Fu. *N*_OH_ corresponds to the number of reactive −OH groups per unit in HA or Fu structure.

The DM of MHA and MFu were determined according to the equation below [28].
(1)$$\mathrm{DM}=\left[\left(I_{\mathrm{methylene}}/N_{\mathrm{methylene}}\right)/\left(I_{\mathrm{CH}3}/N_{\mathrm{CH}3}\right)\right]∕N_{\mathrm{OH}}\;\times 100\%$$

### 2.4. Preparation of the MHA and MHA-MFu Hydrogels

Hydrogel formation was achieved following an adapted protocol from Hachet et al. [37]. MHA was dissolved in phosphate-buffered saline (PBS, Invitrogen, Carlsbad, CA, USA) at 1% *w*/*v* as a precursor solution. MFu (0.5 and 1.0% *w*/*v*) was added to the MHA precursor solution to generate MHA-MFu0.5 and MHA-MFu1.0 precursor solutions, respectively. All gel precursor solutions contained 0.1% of the photoinitiator Irgacure 2959 and were added to 96-well plates and exposed to 365 nm light at 2.6 mW/cm^2^ (UVP CL-1000, Upland, CA, USA) for 7 min to form the hydrogels (Figure 1A). As mentioned above, Fu is a sulfated polysaccharide, and the position of sulfur (S) can indirectly represent the location of Fu. To confirm the existence of Fu in the hydrogel, the atomic composition of S in the MHA-MFu hydrogel was determined by energy-dispersive spectroscopy (EDS) coupled with scanning electron microscopy (SEM) (Hitachi S-3400, Tokyo, Japan). Figure 1B shows that the MHA-MFu hydrogel contained Fu. The hydrogels were prepared in 96-well plates (6.5 mm × 10.0 mm, diameter × height), 100 μL of MHA, MHA-MFu0.5, and MHA-MFu1.0 precursor solutions and photoinitiator were added into per well, and then exposed to 365 nm light to form the hydrogels. Finally, the hydrogels were removed and washed with PBS to remove all residues of non-photo-crosslinked and photoinitiator. The size of each hydrogel (wet state) was approximately 6.5 mm × 3.5 mm (diameter × height).

### 2.5. Mechanical Properties Assay

The hardness and elastic modulus of hydrogels were measured by texture profile analysis (TPA) (TA-TX2 texture analyzer, Stable Micro Systems Ltd., Surrey, UK) [38], and hydrogels were prepared in 96-well plates as mentioned above and removed from the plates before testing. A 75 mm diameter compression plate was used to compress twice the cylindrical shape of hydrogels to 40% of the original height, and TPA test conditions were as follows: 90.0 mm/min pretest speed, 60 mm/min test speed, 90 mm/min post-test speed, 10 mm target mode distance, 49 N trigger force, auto trigger type, and a 200 points/s data acquisition rate. The waiting time between the first and second compression was 5 s. The TPA analysis was complete within 100 s at room temperature. Three repeat measurements were taken for each sample in the same batch, and the value was calculated with the software provided with the instrument [39]. All hydrogels with or without preosteoblastic MG63 cell culture were stored in culture medium at 37 °C, and the hardness and elastic modulus were assayed on Days 0 and 28.

### 2.6. Culture Conditions and Proliferation of Preosteoblastic MG63 Cells on MHA and MHA-MFu Hydrogels

The human preosteoblastic MG63 cell line was purchased from Bioresource Collection and Research Center (BCRC NO. 60279, Hsinchu, Taiwan). MHA, MHA-MFu0.5, and MHA-MFu1.0 hydrogels were prepared in 96-well plates. The hydrogels were seeded with a density of 5 × 10^3^ cells/well (100 μL/well) MG63 cells in minimum essential medium (MEM) supplemented with 10% (*v*/*v*) fetal bovine serum (FBS), 50 μg/mL ascorbic acid, 10 mM β-glycerophosphate and antibiotics (100 U/mL penicillin, and 100 μg/mL streptomycin) at 37 °C, 5% CO_2_. In a 2D plate cell culture, 0.5% and 1.0% *w*/*v* Fu were added to MEM (Fu0.5 and Fu1 groups, respectively) and then cultured under the same conditions. Cell proliferation was tested by MTT (3-(4,5-dimethylthiazol-2-yl)-2,5-diphenyltetrazolium bromide) colorimetric assay on Days 1, 3, and 7.

### 2.7. ALP-Specific Activity

The cell-seed hydrogels were washed with PBS and incubated in a cell lysis buffer solution (1% Triton, X-100, 0.1 M glycine, and 1 mM MgCl_2_ in PBS) for 20 min. 50 μL of the lysate was added with 150 μL of the ALP activity reagent (Randox ALP detection kit, Crumlin, UK) at 37 °C for 1 h, and the absorbance of the solution was measured at a wavelength of 405 nm. The ALP-specific activity was assayed on Days 3, 7, and 14.

### 2.8. Cell Attachment Using Immunofluorescence and SEM

The cell-seeded hydrogels were washed with PBS and fixed with formaldehyde solution (3.6% in 0.02 M PBS) for 10 min, quenched for 10 min twice with 0.3 M glycine in PBS, then permeabilized with 0.1% Triton X-100 for 5 min. The F-actin was stained by Alexa Fluor™ 488 phalloidin (Thermo Scientific, Waltham, MA, USA) for 30 min, and cell nuclei were stained by 100 ng/mL DAPI (Sigma, Saint Louis, MO, USA) for 30 min in PBS. All images were captured using a laser scanning confocal microscope (Leica TCS SP8, Wetzlar, Germany) and merged with ImageJ software (National Institute of Mental Health, Rockville, MD, USA) for qualitative assessment of the cell morphology. The cell-seeded hydrogels were sputter-coated with gold and imaged by SEM (Hitachi S-3400). Cell attachment was assayed on Day 7.

### 2.9. Mineralization Analysis

Mineralization of MG63 cells on the hydrogels was analyzed using alizarin red-S (ARS) dye [40]. The cell-seeded hydrogels were washed with PBS and fixed in formaldehyde for 15 min. Subsequently, the hydrogels were washed with deionized water and stained with ARS solution (2%, pH 4.2) for 15 min. The bound ARS was dissolved in 10% acetic acid, and the concentration of ARS was assessed by measuring absorbance at a wavelength of 430 nm. The calcium deposits were observed from the red color by microscopy (Nikon TS-100, Tokyo, Japan), and ARS staining was assayed on Day 28.

### 2.10. Statistical Analysis

Quantitative data were collected in triplicate (*n* = 3) and are reported as the mean ± standard deviation where indicated. Statistical analyses were performed using SPSS v. 10 (International Business Machines Corporation, Armonk, NY, USA). Values of *p* < 0.05 were considered significant.

## 3. Results and Discussion

### 3.1. Characterization of MHA and MFu

The raw spectra of the HA and MHA samples are shown in Figure 2A. The MHA spectrum revealed the appearance of a characteristic ester peak (C=O) at 1723 cm^−1^, which is present in the methacrylate group but not in HA, thus confirming the methacrylation of MHA.

The DM of MHA was calculated by ^1^H NMR from the presence of two groups of peaks; one at δ = 6.21 and 5.78 ppm, which was due to the methylene (=CH_2_; H_a_ and H_b_) in the double bond region of MA, and the other at δ = 2.05 ppm, which corresponds to the methyl (CH_3_; H_c_) of HA (Figure 2B). From these data, it was determined that the DM of MHA was 9.2%. The infrared spectra in Figure 3A show signals in the range of 1005–1253 cm^−1^ corresponding to the stretching vibration of the sulfur-oxygen double bond (S=O) [30] of the sulfate group in both Fu and MFu. MFu also had the characteristic ester peak (C=O) at 1723 cm^−1^, which is present in the methacrylate group but not in Fu. Methylene protons (=CH_2_; H_a_ and H_b_) on MA were found at δ = 6.27 and 5.80 ppm, and the δ = 1.36 ppm peak corresponded to the methyl group (CH_3_; H_c_) of Fu (Figure 3B). From these data, it was determined that the DM of MFu was 98.6%. Successful methacrylation of MHA and MFu was confirmed by the presence of ester peaks (C=O) by FTIR and methylene protons (=CH_2_) by ^1^H NMR analyses. In addition, there was a signal appearing at δ = 1.98 ppm in Figure 2B and Figure 3B, which corresponds to the methyl (CH_3_) of MA [36].

### 3.2. Effect of Osteoblast Cells on the Mechanical Properties of the MHA and MHA-MFu Hydrogels

The mechanical properties of the matrix environment have been shown to affect cell function and differentiation [41]. Moreover, substrate elasticity is a potent regulator of osteoblast functionalization, which may pave the way for further understanding bone diseases and be a potential therapeutic alternative for tissue regeneration [42]. Therefore, we first analyzed the mechanical properties of the hydrogels with different components, after different storage times and in the presence or absence of cell attachment. The MHA, MHA-MFu0.5, and MHA-MFu1.0 hydrogels were formed by photopolymerization at 365 nm from precursor solutions in the presence of the photoinitiator Irgacure 2959, which was selected for its known biocompatibility with cells [43].

In our study, the hardness of the hydrogels showed no change after the addition of MFu, and there was no significant difference in hardness after 28 days of storage (Figure 4A). However, the elastic modulus after MFu addition showed a significant decrease compared to that with MHA alone. The elastic moduli of the MHA, MHA-MFu0.5, and MHA-MFu1.0 hydrogels were 107.3 ± 12.2 kPa, 74.5 ± 7.8 kPa, and 66.8 ± 2.8 kPa on Day 0, respectively. Additionally, the hardness and elastic modulus of the hydrogels were not significantly different between Days 0 and 28 (Figure 4B). Xue et al. [44] demonstrated that hydrogels composed completely of high-molecular-weight HA have the highest hardness and elastic modulus values, and the elastic modulus decreases when low-molecular-weight HA is mixed with high-molecular-weight HA. The molecular weights of HA and Fu used in this experiment were 620–1200 KDa and 35–40 KDa, respectively, and the MHA-MFu hydrogels were composed of two different molecular weight polysaccharides. We report similar results to those of Xue et al. [44] for the MHA-MFu0.5 and MHA-MFu1.0 hydrogels, so the elastic moduli of the MHA-MFu0.5 and MHA-MFu1.0 hydrogels are lower than that of the MHA hydrogel. Current literature has reported that the enzymes and metabolites produced when cells interact with hydrogels may affect their mechanical properties [45]. Therefore, next, preosteoblastic MG63 cells were seeded on hydrogels for 28 days to measure the mechanical property changes. The results showed that seeding preosteoblastic MG63 cells on hydrogels did not affect the hardness or elastic moduli of the hydrogels on Days 0 and 28. There was no significant difference between the samples with or without preosteoblastic MG63 cells on Day 28 (Figure 5A,B). Our data demonstrated that MHA-MFu0.5 and MHA-MFu1.0 hydrogels could maintain scaffold stability to bear loads and fulfill defects.

### 3.3. Effects of Fu on Osteoblast Cells in Plate Culture

In this study, to identify the beneficial effects of Fu in promoting osteogenesis in preosteoblastic MG63 cells, we examined whether Fu can affect cell viability and ALP-specific activity and observed the cell morphology under a microscope. On all experimental days, the addition of Fu decreased cell viability. The cell viability in the Fu0.5 group was higher than that in the Fu1 group, and there was a statistically significant difference (Figure 6A). It has been suggested that a high concentration of Fu would inhibit the viability of preosteoblastic cells under plate culture [46]. The effect of Fu on the maturation of preosteoblasts to osteoblasts was studied by determining ALP-specific activity [47]. Our results showed that Fu increased ALP-specific activity after three days (Figure 6B) and was not affected by the inhibition of cell viability. Cho et al. [48] also reported that Fu can increase ALP-specific activity as a phenotypic marker for early-stage osteoblastic differentiation of MG63 cells. To investigate the effects of Fu on MG63 cells in detail, we examined changes in cell morphology after the cells were treated with Fu for seven days. Cells that were in the presence of 0.5% and 1% Fu (Fu0.5 and Fu1 groups, respectively) were rounder than those in the control group (Figure 6C). Fu has been reported to induce the accumulation of F-actin in the cell cortex via the PI3K and Akt signaling pathways [26,49,50]. Our results suggested that Fu decreased cell viability and promoted ALP-specific activity to induce MG63 differentiation. Thus, Fu might compensate for the defects of HA that do not support cell mineralization

### 3.4. Cell Behavior on MHA and MHA-MFu Hydrogels

The spreading of attached osteoblasts was also dependent on the chemistry of the hydrogel surface. Lam et al. found that HA hydrogels showed that RGD peptides promoted spreading and motility of cells [51] Li et al. identified that decoration of the bioactive motif in HA hydrogels could promote cell adhesion and osteogenic differentiation [52]. In this study, preosteoblastic MG63 cells adhered to the MHA, MHA-MFu0.5, and MHA-MFu1.0 hydrogels were stained with Alexa Fluor™ 488 phalloidin to detect the actin cytoskeleton and with DAPI to detect nuclei and were observed under a confocal laser scanning microscope. The cells on the MHA-MFu0.5 and MHA-MFu1.0 hydrogels showed spherical morphologies with very low cell spreading that was similar to the cells attached to the MHA hydrogel (Figure 7). This indicated that the incorporation of Fu did not alter the cell morphology within the hydrogel. Studies have reported that cell shape can regulate cell growth [53], gene expression [54], and ECM metabolism [55]. Moreover, our SEM analysis results showed that the MG63 cells adhered to the MHA hydrogel had smooth surfaces. However, the surface and the surroundings of the cells were rough in the MHA-MFu0.5 and MHA-MFu1.0 hydrogel groups (Figure 7). Karunanithi et al. [27] reported that this difference in the cell surface and surroundings may arise from a higher production of ECM-related components when cells are attached to Fu-containing hydrogels. Our results suggested that MHA-MFu0.5 and MHA-MFu1.0 hydrogel incorporation might support ECM formation in MG63 cells.

The molecular composition of a hydrogel can affect cell behavior, such as attachment, proliferation, and differentiation [56]. The effects of Fu in a hydrogel on cell attachment, viability, and differentiation were examined by an MTT assay and an ALP-specific activity assay. Quantitative measurements of cell viability on MHA, MHA-MFu0.5, and MHA-MFu1.0 hydrogels at one, three, and seven days showed that the cell viabilities in the three groups were not significantly different after seven days (Figure 8A). ALP-specific activity, an early marker of the osteoblastic phenotype, was higher in the MHA-MFu1.0 hydrogel group than in the MHA and MHA-MFu0.5 hydrogel groups at 7 and 14 days (Figure 8B). The above results indicated that Fu inhibited cell viability when directly added to the medium (Figure 6A). However, Fu modified with methacrylate and prepared into a hydrogel had no significant effect on cell viability (Figure 8A) and still promoted ALP-specific activity (Figure 8B) as was observed in medium (Figure 6B). The purpose of this study was to provide a scaffold that can maintain flexibility during brain swelling and then turn into bone after swelling to implement a single-stage surgery to treat TBI. Finally, we analyzed the mineralization ability of MG63 cells on the hydrogels by colorimetric calcium quantification. Mineralization was significantly higher in the MHA-MFu0.5 and MHA-MFu1.0 hydrogel groups than in the MHA hydrogel group after 28 days. In addition, the level of mineralization increased as the concentration of Fu in the hydrogel increased (Figure 9). These results indicate that the MHA-MFu1.0 hydrogel promotes osteoblast differentiation and activity.

## 4. Conclusions

In conclusion, we demonstrated that the MHA-MFu hydrogels enhanced osteoblast differentiation, probably due to stimulation of ALP-specific activity and mineralization. This marine-sourced sulfated polysaccharide can be found in nature and is easily available, so it can be considered a promising additive to promote effective and reliable bone graft materials. Altogether, our results suggested that Fu could be used as a potential candidate to form a biocomposite scaffold for bone tissue engineering applications because it allows attached cells to differentiate and mineralize.

## Figures and Tables

**Figure 1 materials-14-01168-f001:**
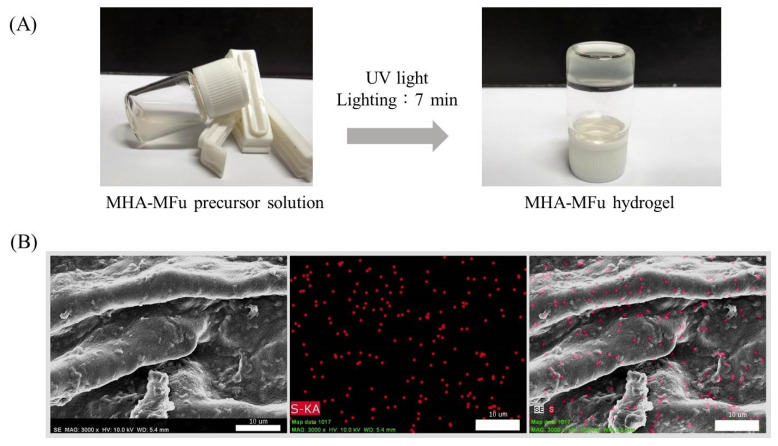
Hydrogel formation (**A**) and SEM images of elemental mapping of sulfur (S) in the MHA-MFu hydrogel (**B**). Scale bar = 10 μm.

**Figure 2 materials-14-01168-f002:**
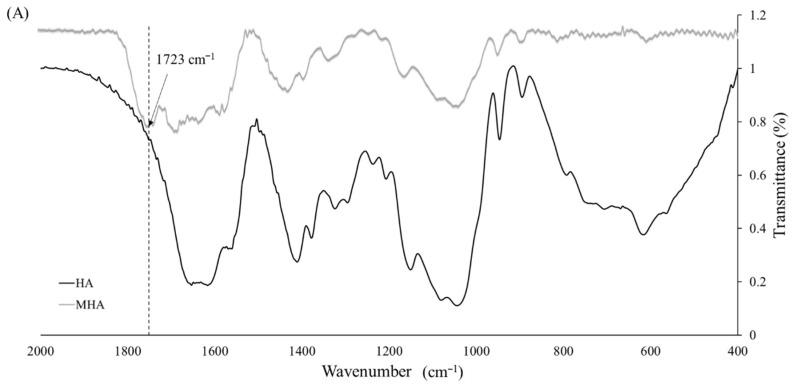
FTIR spectra (**A**) and ^1^H NMR spectra (**B**) of hyaluronan (HA) and methacrylate-hyaluronan (MHA).

**Figure 3 materials-14-01168-f003:**
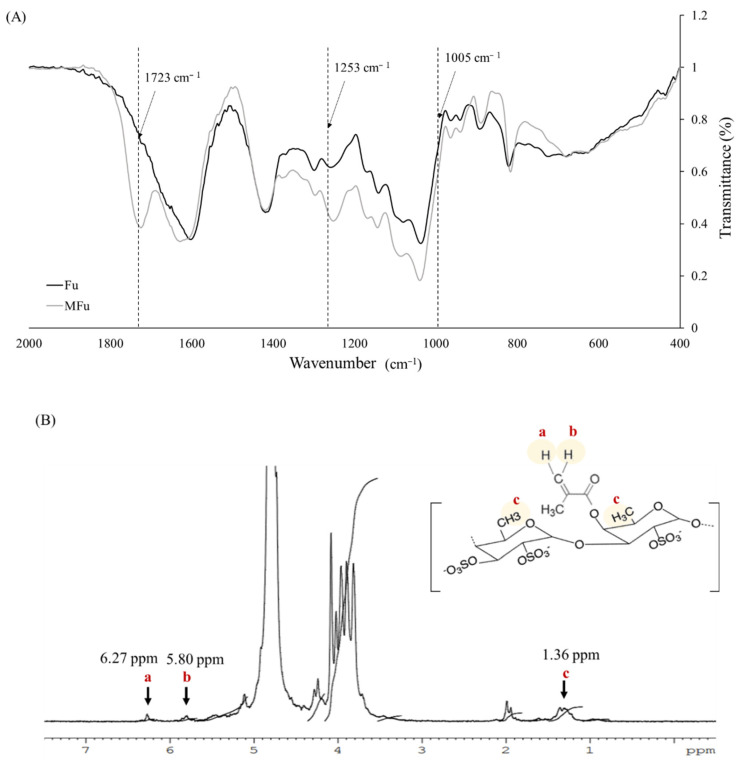
FTIR spectra (**A**) and ^1^H NMR spectra (**B**) of fucoidan (Fu) and methacrylate-fucoidan (MFu).

**Figure 4 materials-14-01168-f004:**
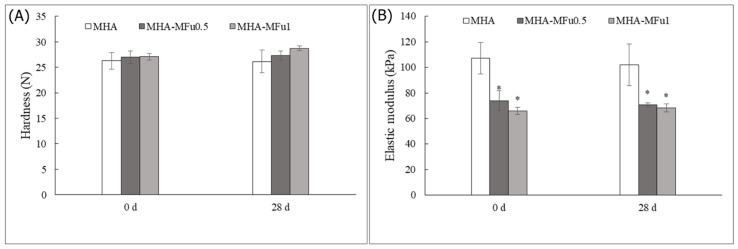
The mechanical properties of the MHA and MHA-MFu hydrogels: (**A**) hardness and (**B**) elastic modulus after 28 days. * (*p* < 0.05) indicates the level of significance compared with MHA.

**Figure 5 materials-14-01168-f005:**
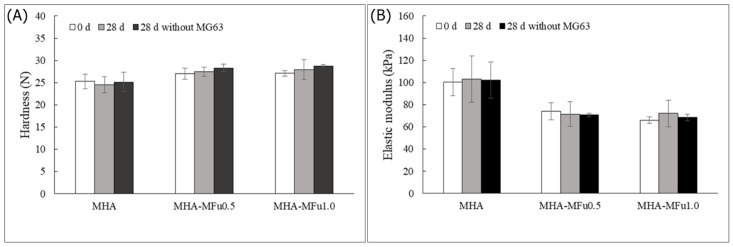
The mechanical properties of the MHA and MHA-MFu hydrogel formulations (**A**) hardness and (**B**) elastic modulus with MG63 cells after 28 days. There was no significant difference between 0 d and 28 d with cells and 28 d without cells.

**Figure 6 materials-14-01168-f006:**
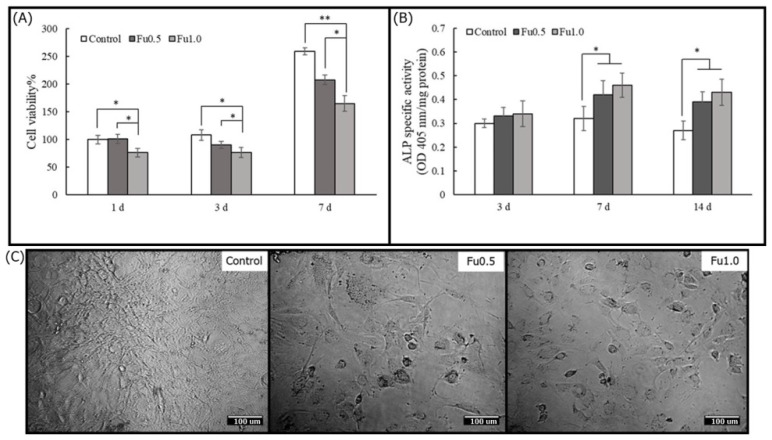
The cell viabilities (**A**) and ALP-specific activities (**B**) of Fu with MG63 cells in plate culture on Days 1, 3, and 7 and the cell morphology on Day 7 (**C**). The normalization was based on the viability of cells cultured on MEM medium (control). * (*p* < 0.05) and ** (*p* < 0.01) indicate the level of significance. Scale bar = 100 μm.

**Figure 7 materials-14-01168-f007:**
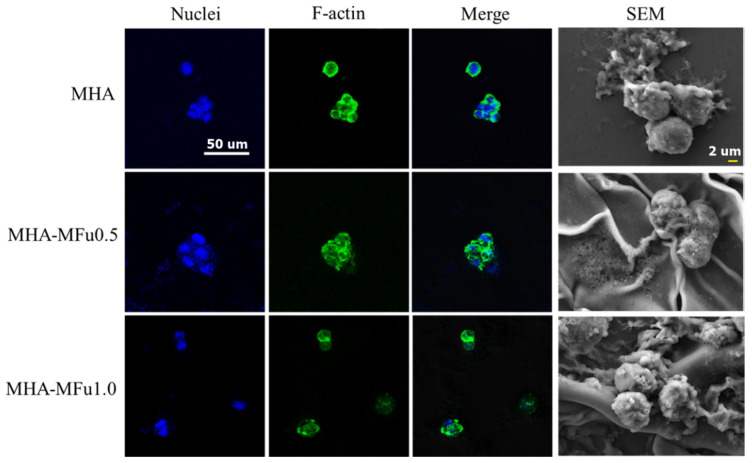
Immunofluorescence and SEM analysis of cell morphology on the MHA and MHA-MFu hydrogels after 7 days. MG63 osteoblasts were stained with F-actin (green), and cell nuclei were stained with DAPI (blue). Scale bar = 50 μm for immunofluorescence images and 2 μm for SEM images.

**Figure 8 materials-14-01168-f008:**
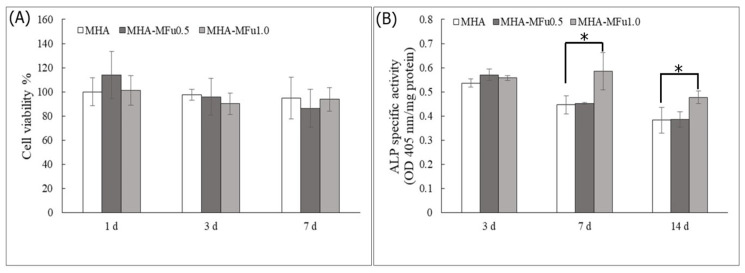
The cell viabilities (**A**) and ALP-specific activities (**B**) of MG63 osteoblasts on MHA and MHA-MFu hydrogels. The normalization was based on the viability of cells cultured onto MHA. * (*p* < 0.05) indicates the level of significance.

**Figure 9 materials-14-01168-f009:**
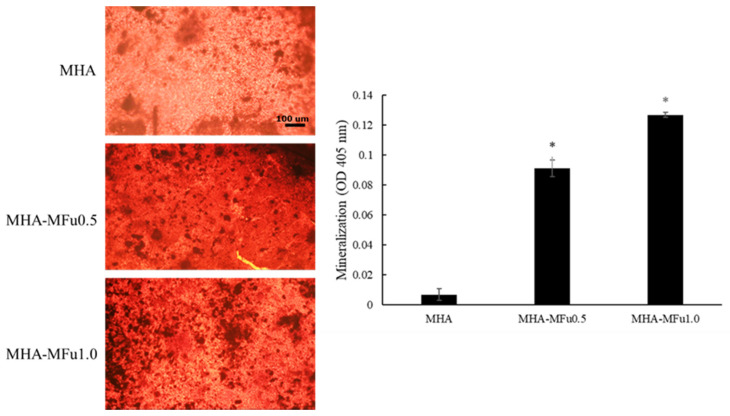
Bone mineralization of MG63 osteoblasts on MHA and MHA-MFu hydrogels after 28 days. * (*p* < 0.05) indicates the level of significance. Scale bar = 100 μm.

## Data Availability

Data is contained within the article.

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
