# Peer review of "Preparation of a Fucoidan-Grafted Hyaluronan Composite Hydrogel for the Induction of Osteoblast Differentiation in Osteoblast-Like Cells"

_materials, 2021, doi:10.3390/ma14051168_

Round 1

Reviewer 1 Report

The present manuscript describes the preparation of fucoidan-mixed hyaluronan composite hydrogel for potential use in bone tissue engineering. The objective seems interesting with potential application in biomedical engineering. Besides, the experimental methods have properly described and performed with a good discussion of results. However, revision is required and some comments should be taken into account:

- I miss a comparison of fucoidan with other sulfated glycosaminoglycans, for example, chondroitin sulfate.

- Hydrogels are characterized by a high degree of hydrophilicity. How does the functionalization by methacrylic groups affect hydrophilicity and, as a consequence, mechanical properties of the final hydrogels? Please, include mechanical properties of HA hydrogels in Figure 4 and 5 in order to compare.

- Was the degree of methacrylation (DM) calculated from the FTIR or NMR measurements? In addition, a relation between DM and hydrophilicity would be welcome.

- Why have TPA measurements been carried out at 25 ºC? How does temperature affect the behaviour of these hydrogels? What about 37 ºC? Furthermore, I think that different dynamic rheological measurements would be interesting to evaluate the mechanical behaviour of the gel.

- I consider that different techniques have been performed in order to characterize in vitro behaviour, but I miss in vivo assessment.

Author Response

Thanks very much for your comments, and the point-by-point response was attached as a word file.

Reviewer 2 Report

This paper is focused on the preparation and characterization of a grafted fucoidan-hyaluronan biocomposite hydrogel scaffold suitable for bone tissue engineering applications. From this point of view, the present paper is of some interest and can be considered for publication in Materials. However, the material part must be described and presented in a reliable and accurate manner allowing others to repeat the presented research.

Specific comments:

  1. Line 97: The properties of hyaluronic acid are very dependent on the molecular weight; therefore, the wide ranges of values provided by the manufacturer (620-1200 kDa) are clearly insufficient. I recommend that the authors determine the molecular weight of HA by any available method (viscometry, light scattering, or size exclusion chromatography).
  2. Lines 100-102: Specify methods for determining MW, monomeric composition, and sulfate content of fucoidan and give appropriate references.
  3. Line 133: Hyaluronic acid contains one methyl group per 2 monomeric units, whereas fucoidan contains a methyl group in each monomeric unit. Therefore, the formula for calculating the degree of methacrylation cannot be the same for both polysaccharides. Correct the equation and the degree of metacrylation if necessary.
  4. Section 2.3: It would also be advisable to use the signals of the methyl group of the methacrylate moiety to calculate the degree of methacrylation for both MHA and MFu.
  5. Figures 1,6,7,9: Please indicate clearly the scale bars. In most cases, they are barely visible.
  6. Figure 2: Clearly indicate the signal of the methyl group of HA (you can make an inset of this part of the spectrum with a larger scale). Discuss in the text which protons correspond to two remaining signals at around 1.9 and 2.0 ppm.
  7. Figure 3: (c) box is in the wrong place. Please move it to re right place at around 1.3 ppm. Also, assign and discuss in the signal at around 2.0 ppm.
  8. Line 81: Consider citing here the recent papers on this topic (DOI: 10.3390/biology10010067, 10.1016/j.bcdf.2019.100209).

Author Response

(The authors gave the same response as above.)

Round 2

Reviewer 1 Report

In the revised manuscript, authors made some necessary revision and basically addressed the responses to the questions my comments. The paper was generally improved and, in my opinion, can be published in present form.

Author Response

We are very grateful for your suggestions to make our report more complete.

Reviewer 2 Report

The authors have successfully addressed most of the reviewers’ concerns, improving the manuscript with their edits. However, there are still a couple of minor points for authors to consider:

  1. The degree of substitution (methacrylation in this case) is commonly expressed per monomeric unit of a polysaccharide rather than per dimeric unit. The way of expressing the degree of substitution to the dimeric unit is particularly unsuitable for fucoidan, as the ratio of α-1→3- and α-1→4-linked fucose residues (with different functionalities and reactivities) varies from sample to sample. Please recalculate DM given 1.5H/unit for hyaluronan and 3H/ unit for fucoidan.
  2. Line 111: What do you mean by the 47.5±0.9% mol fucose content of the sample? What is the remaining 52.5%?
